# Status of Cassava Witches’ Broom Disease in the Philippines and Identification of Potential Pathogens by Metagenomic Analysis

**DOI:** 10.3390/biology13070522

**Published:** 2024-07-15

**Authors:** Darwin Magsino Landicho, Ray Jerome Mojica Montañez, Maurizio Camagna, Sokty Neang, Abriel Salaria Bulasag, Peter Magan Magdaraog, Ikuo Sato, Daigo Takemoto, Kensaku Maejima, Marita Sanfuego Pinili, Sotaro Chiba

**Affiliations:** 1Central Laboratory, National Plant Quarantine Services Division, Bureau of Plant Industry, Manila 1004, Philippines; 2Nagoya University Asian Satellite Campuses Institute, Philippine Campus, University of the Philippines Los Baños, Laguna 4031, Philippines; 3Graduate School of Bioagricultural Sciences, Nagoya University, Nagoya 464-8601, Japansoktyneang@gmail.com (S.N.); asbulasag@up.edu.ph (A.S.B.); isato@agr.nagoya-u.ac.jp (I.S.); dtakemo@agr.nagoya-u.ac.jp (D.T.); 4Central Post-Entry Quarantine Station , Bureau of Plant Industry, Los Baños, Laguna 4030, Philippines; rayjeromemontanez@gmail.com; 5College of Arts and Sciences, University of the Philippines Los Baños, Laguna 4031, Philippines; 6Crop Pest Management Division, Bureau of Plant Industry, Manila 1004, Philippines; cpmd@buplant.da.gov.ph; 7Biology Department, College of Science, De La Salle University, Manila 0922, Philippines; 8Graduate School of Agricultural and Life Sciences, The University of Tokyo, Tokyo 113-8657, Japan; amaejima@mail.ecc.u-tokyo.ac.jp; 9National Crop Protection Center, College of Agriculture and Food Science, University of the Philippines Los Baños, Laguna 4031, Philippines; mspinili@up.edu.ph

**Keywords:** CWBD, *Candidatus* Phytoplasma luffae, nested PCR, next-generation sequencing, etiology, *Ceratobasidium*

## Abstract

**Simple Summary:**

Cassava witches’ broom disease (CWBD) is a major threat to cassava production, and it is believed to be caused by phytoplasma, an unculturable bacterial pathogen. However, recent findings suggest that other pathogens, such as viruses or fungi, may cause witches’ broom disease in other crops. This study aims to investigate CWBD in the Philippines, specifically its status, and identify its potential pathogen(s). Currently, CWBD has spread nationwide and causes severe symptoms, such as leaf clustering and drying, as well as the browning of stems and roots. Polymerase chain reaction (PCR) and next-generation sequencing (NGS) showed consistent detection and a high abundance of *Ceratobasidium* sp., but an inconsistent and low detection rate of phytoplasma (*Candidatus* Phytoplasma luffae) in Philippine cassava samples. These findings challenge existing notions about CWBD and its causal agent. Confirming the causality and understanding *Ceratobasidium* sp. involvement can lead to better disease diagnosis and control strategies for ensuring food security.

**Abstract:**

Cassava witches’ broom disease (CWBD) is one of the most devastating diseases of cassava (*Manihot esculenta* Crantz), and it threatens global production of the crop. In 2017, a phytoplasma, *Candidatus* Phytoplasma luffae (*Ca.* P. luffae), was reported in the Philippines, and it has been considered as the causal agent, despite unknown etiology and transmission of CWBD. In this study, the nationwide occurrence of CWBD was assessed, and detection of CWBD’s pathogen was attempted using polymerase chain reaction (PCR) and next-generation sequencing (NGS) techniques. The results showed that CWBD has spread and become severe, exhibiting symptoms such as small leaf proliferation, shortened internodes, and vascular necrosis. PCR analysis revealed a low phytoplasma detection rate, possibly due to low titer, uneven distribution, or absence in the CWBD-symptomatic cassava. In addition, NGS techniques confirm the PCR results, revealing the absence or extremely low phytoplasma read counts, but a surprisingly high abundance of fastidious and xylem-limited fungus, *Ceratobasidium* sp. in CWBD-symptomatic plants. These findings cast doubt over the involvement of phytoplasma in CWBD and instead highlight the potential association of *Ceratobasidium* sp*.,* strongly supporting the recent findings in mainland Southeast Asia. Further investigations are needed to verify the etiology of CWBD and identify infection mechanisms of *Ceratobasidium* sp. to develop effective diagnostic and control methods for disease management.

## 1. Introduction

Cassava (*Manihot esculenta* Crantz) is a valuable root crop [1]. According to the Food and Agriculture Organization (FAO), cassava is the third most important calorie source in the tropics, next to rice and corn [2]. In 2021, 314 million tons of cassava were produced globally [3]. It is the major food source for over 500 million people worldwide [4,5]. In the Philippines, cassava is the most widely propagated root crop grown mainly for food, feed, fuel, and raw materials for industrial products [6,7]. Predominantly cultivated in small-holder farms, its ability to grow in harsh conditions appeals to poor and resource-constrained farmers who rely on cassava for food and a source of livelihood [8]. Thus, developing the cassava industry impacts poverty alleviation and food security [9,10]. In recent years, the Philippines has been expanding its cassava industry through concerted efforts from the government and private institutions. As a result, there has been an increasing intensification of cassava production in recent decades due to an increase in demand for cassava for food and feed [9]. However, this intensification also resulted in the widespread movement of cassava planting materials and, in turn, emerging pests and diseases [10,11].

The expansion of the cassava industry has been threatened by the occurrence of cassava witches’ broom disease (CWBD). Similar symptoms, described as *Superbrotamento* or *Envassouramento*, were observed in Brazil in the 1940s [12] and then reported in Uganda in 2009 [13]. In Southeast Asia including the Philippines, witches’ broom symptoms were already recognized in 2012 [14,15]. CWBD causes yellowing, clustering of axillary buds and lateral shoots or “witches’ broom”, vascular necrosis, and stunting [4]. Based on recent reports, CWBD can cause yield loss of up to 50% by reduced root volume and starch content, and an increase in cyanide content [16].

CWBD is believed to be caused by a phytoplasma, an obligated phloem-colonizing bacteria lacking cell walls [17,18,19]. It is generally transmitted by insect vectors such as leafhoppers, plant hoppers, and psyllids and carried over long distances by human-mediated transport of infected planting materials [11]. Due to its obligate nature, phytoplasma detection and identification relies on the use of molecular diagnostic tools such as polymerase chain reaction (PCR) and sequencing [17]. However, it is often challenging due to low titer and uneven distribution of phytoplasma in the host plant [20]. Identification of phytoplasma typically relies on the sequencing of the 16S rRNA gene amplified by nested PCR [21,22]. Previously, different groups of phytoplasma have been detected from CWBD-infected plants in different countries, such as 16S rDNA sequence-based taxonomic groups 16Sr-I, II, V, VI, VIII, XII, and XV, respectively [13,23,24,25,26,27,28]. In addition to cassava, phytoplasma has been detected in bitter gourd (*Momordica charantia*), sponge gourd (*Luffa aegyptica*), bamboo (*Bambusa vulgaris*), and rice (*Oryza sativa*) in the Philippines [29,30,31]. Most importantly, the CWBD etiology has never been established due to the lack of definitive studies addressing the pathogenicity of phytoplasma in cassava to fulfill Kochs’ postulate.

In Southeast Asia, CWBD was also attributed to phytoplasma infection. However, recent studies have reported non-detection of phytoplasma in symptomatic leaves and the difference in the symptoms observed in the Americas such as branching at the base of the main stem accompanied by narrowing of leaves [14,23,32,33]. A significant development in CWBD research was recently reported by Leiva et al., where phytoplasma was not detected and the fungus *Ceratobasidium* sp. was discovered using a metagenomic approach and found associated with CWBD using a set of specific PCR primers [34]. This questions the current notions of phytoplasma as the potential causal pathogen of CWBD in Southeast Asia, and possibly in the Philippines

Due to the difficulties in phytoplasma detection using PCR, the use of next-generation sequencing (NGS) technology offers an alternative approach for assessing microbial composition in phytoplasma-infected plants, as demonstrated by recent studies [35,36,37,38,39]. NGS can be a useful tool to perform culture-independent, high-throughput profiling of microbial communities in diseased tissues. In particular, it is useful for identifying potential fungal, bacterial, and viral pathogens associated with symptomatic crops by assessing their relative abundance [40,41,42,43].

In this study, we report the latest disease status of CWBD in the Philippines and our attempts to detect the potential pathogens using PCR and NGS techniques. We also aim to contribute to the current understanding of phytoplasma and *Ceratobasidium* dynamics in CWBD. Our findings revealed microbial compositions in CWBD-affected cassava leaves, highlighting the relative abundance of *Ceratobasidium* and low detection rate of phytoplasma, which strongly supports the recent findings in Southeast Asia [34].

## 2. Materials and Methods

### 2.1. Assessment of Disease Incidence and Severity

The incidence and severity of CWBD in the Philippines were assessed using field observations and analysis of secondary data from the Bureau of Plant Industry, Crop Pest Management Division (BPI-CPMD). The surveillance data contains CWBD incidence and severity ratings of randomly selected 50 plants per site in major cassava-producing regions, using the established survey protocol of the BPI-CPMD. The incidence and symptom severity records from 2020 to 2022 of the provinces of Isabela, Quirino, Laguna, Quezon, Leyte, Southern Leyte, Eastern Samar, Cebu, Bohol, Negros Oriental, Cotabato, Misamis Oriental, Bukidnon, South Cotabato, and Zamboanga del Norte from seven regions (Regions II, IV-A, VIII, BARMM, X, XII, and IX in the national administrative division) were used for the analysis. The disease severity was assessed using the disease rating scale developed by the National Cassava Surveillance Project (Appendix A). The maximum recorded severity was mapped using QGIS version 3.26.3-Buenos Aires, and the monthly incidence rate was computed based on the survey data.

### 2.2. Sample Collection

Cassava plants showing CWBD symptoms such as witches’ broom and shortened internodes in the provinces of Isabela and Bukidnon were randomly sampled from 2022 to 2023 for PCR and NGS analyses. Twenty leaf tissues and stem cuttings were obtained for nucleic acid extraction and propagation in the screen house with permission from local authorities.

Screen-house-propagated 1-month-old and field-collected 3-month-old cassava (variety Rayong 72) that showed early and advanced witches’ broom symptoms, respectively, were sampled from Bukidnon and Isabela provinces and subjected to microbial RNA Seq analysis. The 1-month and 3-month-old stages were chosen because witches’ broom symptoms manifested earliest during these stages, and younger tissues typically yield higher quality RNA due to lower levels of polysaccharides and secondary metabolites [44]. Asymptomatic cassava with the same age, variety, and location was also sampled and included in the analysis. Leaf samples were surface sterilized with 70% ethanol, sterile distilled water, and blot-dried with sterile tissue papers. Leaf samples from the screen house were processed immediately while field-collected samples were placed in sterilized 50 mL tubes and immediately frozen by dry ice to prevent RNA degradation. Upon arrival at the laboratory, the samples were immediately placed in a −80 °C freezer.

For testing of localization and possible co-occurrence of phytoplasma and *Ceratobasidium* sp., an additional 17 CWBD-symptomatic cassava plants from Isabela province planted in an enclosed field at the National Crop Protection Center, University of the Philippines Los Baños (NCPC, UPLB) were randomly sampled at two different stages. Leaves were obtained during the pre-harvest or vegetative stage (7 months) and leaves, stems, and roots were collected during the harvesting stage (9 months). Several tissue samples from the top, middle, and bottom portions of the plant (2 to 3 samples per plant) were obtained during the pre-harvesting stage, and leaves, stems, and root tissues from the same plant were obtained during the harvesting stage. In addition, 50 leaf samples from CWBD-asymptomatic 9-month-old cassava plants were collected from Los Baños, Laguna as negative (healthy) control. All the samples were placed in a clean sample bag and immediately processed or stored in the −80 °C freezer.

### 2.3. DNA Extraction and Microbial DNA Enrichment

Leaf midrib, petiole, and vascular tissues were selected for nucleic acid extraction to maximize the recovery of the pathogen. Total genomic DNA of symptomatic plants was extracted using a modified DNeasy^®^ Plant Mini Kit (Qiagen^®^, Hilden, Germany) procedure [45]. Tissues were homogenized using a sterilized mortar and pestle under the presence of liquid nitrogen. Homogenized samples were lysed using 2.5% CTAB (1 mL/g of tissue) followed by spin-column purification. The modified protocol is suitable for starchy and woody cassava tissues and reported in phytoplasma diagnostic protocols [45,46,47].

In preparation of samples for NGS, high-quality genomic DNA was enriched for microbial DNA using NEBNext^®^ Microbiome DNA Enrichment kit (New England Biolabs^®^, Ipswich, MA, USA). It has been reported to deplete plant host DNA while enriching non-methylated DNA from phytoplasmas, other bacteria, fungi, and various eukaryotic microbes [48]. A total of 1 μg of DNA was used for the enrichment.

### 2.4. RNA Extraction

To prevent RNA degradation, handling samples under liquid nitrogen after removal from the −80 °C freezer was ensured to prevent thawing. Before extraction, all the reagents used were RNA-grade, sterilized, DEPC-treated, and surfaces decontaminated using RNase AWAY™ (ThermoFisher Scientific, Waltham, MA, USA). Total RNA was extracted on the screen house-collected samples using Qiagen^®^ RNeasy Plant Mini Kit (Qiagen^®^, Hilden, Germany) with minor modifications, particularly increasing washing steps as indicated by previous studies [49,50]. A modified SDS-LiCl method was employed for the extraction of RNA from field-collected leaf samples (3 months) due to the challenges encountered when using an RNA extraction kit with severely infected tissues [44]. Leaf samples were homogenized using sterilized DEPC-treated mortar and pestle under constant application of liquid nitrogen to prevent thawing. Homogenized samples were lysed using RNA extraction buffer (600 µL per 0.1 g tissue) and 2% SDS. Phase separation was performed using acidic phenol chloroform isoamyl alcohol (25:24:1) and precipitation by 2 M Lithium chloride (final concentration) and 1/10 volumes of 3 M Sodium acetate pH 4.8. Incubation was performed overnight at −20 °C and with RNA-grade glycogen to enhance precipitation (ThermoFisher Scientific, Waltham, MA, USA). All centrifugations were performed at 15,890× *g* for 5 min at 4 °C. Dried pellets were dissolved in DEPC water and treated with DNase (Promega, WI, USA). RNA samples were stored in a −80 °C freezer.

RNA quality and quantity were analyzed using NanoDrop™ Spectrophotometer 2000 (ThermoFisher Scientific, Waltham, MA, USA) and QuantiFluor^®^ RNA System (Promega, Madison, WI, USA) in Quantus™ Fluorometer (Promega, Madison, WI, USA) following the manufacturer’s protocol. RNA quality was visualized using 1% ultrapure agarose gel (Vivantis, Selangor, Malaysia) in 1× Tris-acetate-EDTA (TAE) buffer (Sigma-Aldrich^®^, St. Louis, MO, USA) and viewed under E-box gel documentation system (Vilber, Marne-la-Vallée, Paris, France).

### 2.5. Polymerase Chain Reaction and Sanger Sequencing

Total genomic DNA was used as a template for nested PCR amplification to detect phytoplasma using universal primers Plant host housekeeping gene COX1 was amplified using real-time PCR to check DNA yield [51,52]. Universal phytoplasma primer pairs targeting 16S rDNA, P1/P7 [53,54] and R16mF2n/R16mR1 [22] were used for phytoplasma detection (Appendix A). PCR amplifications were performed in a thermal cycler (VeritiPro™ Thermal Cycler, Applied Biosystems, Waltham, MA, USA). The 16S rDNA PCR amplicons were then digested with *Sca*I (*Sca*I-HF^®^, New England Biolabs). *Sca*I recognizes the sequence 5′-AGT/ACT-3′ of phytoplasma 16S rDNA and generates blunt-end fragments.

A commercially available phytoplasma-positive PCR control, *Ca*. P. mali belonging to 16SrX (Catalog No. 08009PC, Loewe^®^, Madrid, Spain), was used for checking PCR amplification and sequencing. Other positive controls include previously verified phytoplasma-positive sponge gourd and bitter gourd (*Ca*. P. luffae) [30,31] and asymptomatic cassava (*Ca*. P. pruni) [55]. For negative controls, DNA from healthy cassava, extraction controls, no template controls, and commercially available phytoplasma-negative PCR control (Catalog No. 08009NC, Loewe^®^, Madrid, Spain) were used.

Fungal detection was performed by amplifying the nuclear ribosomal internal transcribed spacer (ITS) using primers ITS5 and ITS4 (Appendix A) [56]. DNA samples obtained from leaf and stem tissues of CWBD-symptomatic and asymptomatic cassava were tested for fungal detection. Using the same DNA samples from CWBD-symptomatic and asymptomatic cassava used for nested PCR targeting phytoplasma, a specific primer based on the 28S sequence was designed to check for co-occurrence and localization of the *Ceratobasidium* sp. in cassava (Appendix A).

All PCR products and *Sca*I-digested amplicons were visualized in 1% ultrapure agarose gel (Vivantis, Selangor, Malaysia) in 1× TAE buffer (Sigma-Aldrich^®^, St. Louis, MO, USA) and ran on gel electrophoresis system (Mupid^®^ One Electrophoresis System, Tokyo, Japan). Nucleic acids were stained with GelRed^®^ nucleic acid stain (Biotium, Fremont, CA, USA) viewed under the E-box gel documentation system (Vilber, Marne-la-Vallée, France) (Supplementary Information). Representative PCR amplicons of nested and end-point PCRs were bidirectional Sanger sequenced by Macrogen© (Seoul, Republic of Korea).

### 2.6. Next-Generation Sequencing

Profiling of microbial composition (i.e., bacteria, fungi, and virus) in CWBD-symptomatic plants was performed using targeted and non-targeted next-generation sequencing techniques such as 16S amplicon sequencing, shotgun metagenomic sequencing, and microbial RNA-Seq. Enriched microbial DNA was prepared from randomly selected nested PCR-negative 6 symptomatic samples, including 4 samples from Bukidnon and 2 samples from Isabela. For bacterial composition profiling, 16S amplicons were sequenced by GikenBio, Japan. This includes amplification, quality checking, library preparation, deep sequencing, and standard data analysis. Briefly, DNA samples were amplified by the sequencing facility using primers 341F and 805R targeting V3–V4 regions of bacterial 16S rRNA genes [57,58]. The samples were pair-end sequenced (300 bp × 2) on a MiSeq sequencing platform using a MiSeq Reagent kit v3, 600 cycles (Illumina^®^, San Diego, CA, USA), following the standard guidelines for preparing and loading samples. To profile other microbes such as fungi and viruses, the same DNA samples used for amplicon sequencing were pooled per location and then sent to Genome Read Co., Ltd., Takamatsu, Japan, for shotgun metagenomic sequencing. Sequencing libraries were prepared using MGIEasy FS DNA Library Prep Set (MGI, Cat. Np 100006987) and enzymatically fragmented, purified, and size selected by the facility. The samples were pair-end sequenced (150 bp × 2) on a DNBSEQ platform (MGI Tech, Shenzhen, China) using the DNBSEQ-T7 High-throughput Sequencing Set.

To compare the fungal and RNA virus compositions in symptomatic and asymptomatic cassava, sequencing of microbial RNA transcripts was performed. Extracted RNA from leaves of 1-month-old screenhouse propagated symptomatic cassava stem from Bukidnon and Isabela (I-A and I-B), and leaves of 3-month-old field collected symptomatic and asymptomatic cassava from Isabela (symptomatic and asymptomatic cassava with 3 biological replicates each and labeled as I-1, I-2, I-3 and H-1, H-2, H-3, respectively) were sequenced by Macrogen©, Korea and Genome Read Co., Ltd., Takamatsu, Japan, respectively. Sequencing includes quality checking and library preparation. For samples sequenced by Macrogen© (I-A and I-B), sequencing libraries were prepared using Illumina^®^, TruSeq Stranded Total RNA with Ribo-Zero Plant to deplete ribosomal RNA. The prepared library was pair-end sequenced (150 bp × 2) on the Illumina^®^ NovaSeq 6000 platform to generate 100 million reads. For samples sequenced by Genome Read Co., Ltd., (I-1, I-2, I-3, H-1, H-2, and H-3), library preparation was performed using QIAseq FastSelect –rRNA Plant Kit and MGIEasy RNA Directional Library Prep Set (MGI, Cat.1000006385). Libraries were pair-end sequenced (150 bp × 2) using DNBSEQ-G400RS High-throughput Sequencing Set in DNBSEQ-G400R by DNA Nano Ball (DNB) method to generate 15 to 18 million reads.

### 2.7. Bioinformatic Analysis

Sanger sequencing results were checked, trimmed, and analyzed using Geneious Prime^®^ 2023 v2.1. Forward and reverse sequences were aligned, and consensus sequences were run in nucleotide BLAST (BLASTn) search to check for similarity or identity to publicly recorded sequences [59]. Phytoplasma presence and taxonomic group assignment were also checked using *i*Phyclassifier http://plantpathology.ba.ars.usda.gov/cgi-bin/resource/iphyclassifier.cgi (accessed on 23 August 2023) for the 16S rRNA. The specificity of the *Sca*I restriction enzyme to cleave the restriction site 5′-AGT/ACT-3′ was verified by analysis of phytoplasma and non-phytoplasma bacterial sequences in silico using Geneious Prime^®^ v2023.2.1 (Appendix A). All 16S rDNA partial sequences generated in this study were used to compare reference sequences of phytoplasma and other bacteria with the highest BLASTn similarity. PCR primers were designed using Geneious Prime^®^ 2023 v2.1 and specificity was checked using PrimerBLAST https://www.ncbi.nlm.nih.gov/tools/primer-blast/index.cgi (accessed on 6 November 2023) [60]. The presence of 16S rRNA chimeric sequences was checked before submission to NCBI using DECIPHER v2.25.4 [61].

Phytoplasma, other bacteria, and fungal sequences were aligned through the process of Multiple Alignment using the Fast Fourier Transform (MAFFT) algorithm. To perform this, we used the MAFFT version 7.490 plugin in Geneious Prime^®^ v2023.2.1. Phylogenetic trees were constructed using the Tamura-Nei (TN) distance model and Neighbor-Joining tree build method with 1000 bootstraps [62,63,64]. TN distance model was used since it considers substitution rate differences of nucleotides and inequality of nucleotide frequencies to correct for multiple hits [64]. *Nostoc* sp. (LC322125.1), *Botryobasidium obtusisporum* (DQ898729.1), and *Botryobasidium botryosum* (DQ267124.1) were used as outgroups for the 16S rDNA, fungal ITS, and 28S phylogenetic trees, respectively [65,66].

For the analysis of 16S amplicon sequencing data, standard data analysis was performed using QIIME 2 v2022.8 [67]. Quality control such as the removal of chimeric and noisy sequences was performed using the DADA2 plugin of QIIME 2. The feature-classifier plugin was used to obtain representative sequences and SILVA v13. Metagenomic reads were visualized using Krona v2.8.1 [68], and relative abundance reads corresponding to bacterial species were displayed in a stacked bar chart. Phytoplasma reads were confirmed using Geneious Prime^®^ v2023.2.1, and nucleotide BLAST (BLASTn). Host reads were filtered by manually removing the amplicon sequence variant (ASV) counts corresponding to mitochondrial and chloroplast sequences.

Shotgun metagenomic reads were classified using Kaiju v1.9.2 against the latest pre-built indexed reference databases containing archaea, bacteria, viruses, fungi, and microbial eukaryotes (nr_euk) [69]. Reads were then de novo*-*assembled using MEGAHIT v1.2.9 in an attempt to generate contigs from phytoplasma or other pathogens [70]. Contigs were aligned against fungal, viral, and bacterial RefSeq databases using local BLAST (BLAST v2.5.0+) to obtain DNA sequences to verify Kaiju taxonomic classification and design primers for PCR amplification. The highest similarity hits and contigs aligned with DNA barcode regions, particularly ITS and 28S were obtained for further analysis.

For microbial RNA-seq data, sequencing adaptors were trimmed using TrimGalore! v0.6.7 [71] and quality was checked using FastQC v0.12.1 [72]. Trimmed reads were mapped to the respective host reference genome, *M. esculenta* (GCF_001659605.2), using HISAT2 v2.2.1 to remove host reads [73]. The unmapped reads were classified by Kaiju v1.9.2 using the same parameters used for genomic reads. Output files were converted to tab-separated tables using kaiju2krona and kaiju2table programs, respectively, and imported in Krona v2.8.1 for visualization. To assess the microbial composition of CWBD-symptomatic and asymptomatic samples, alpha diversity metrics such as Shannon and evenness indexes were computed for bacteria, virus, and fungal communities. Results were plotted and significant differences were calculated using independent *t*-test (*p* < 0.05) using RStudio v 2023.06.0+421. Subsequently, relative abundances were calculated by dividing the taxonomic read count at the genus and family levels by the total read count of each microbial pathogen group such as bacteria, fungi, and viruses. Taxonomic assignments were arranged based on median highest relative abundance and visualized in stacked bar plots. The top ten most abundant taxa, determined by relative abundance, were displayed. Significance testing for differences between symptomatic and asymptomatic groups was conducted using a one-tailed Mann-Whitney U test (*p* < 0.05).

## 3. Results

### 3.1. Symptoms and Prevalence of Cassava Witches’ Broom Disease (CWBD) in the Philippines

Based on field surveys during different crop stages (3, 6, and 12 months), clustering of internodes and excessive lateral branch or leaf proliferation were one of the most apparent and distinguishing symptoms of the CWBD across crop stages (Figure 1A,B). In addition, CWBD-affected cassava plants also showed various leaf symptoms, such as chlorosis (yellowing), accumulation of anthocyanin pigments in the apical leaves, stunting, and necrosis (Figure 1B,C). In the later stages, the CWBD eventually caused leaf senescence, shortened internodes, and dieback symptoms (Figure 1B). A radial dissection of stems revealed browning or necrosis of the vascular tissues, on the middle parts of the stem (Figure 1D). This phenotype was extended to the bottom and underground parts of the stems and roots (Figure 1E). These severe symptoms cause a reduction in starch content and root size, thereby significantly reducing yield and quality [16].

Based on field observations and other survey data obtained from BPI-CPMD, the CWBD incidences in the survey-target area were plotted on the map (Figure 2). In major cassava-producing areas in the Philippines, almost all surveyed provinces recognized the occurrence of CWBD. This survey also recorded severities of CWBD-associated symptoms according to a rating sheet (Appendix A), and the highest severities in each plot are illustrated in Figure 2. Accordingly, most plots have an increased severity during the survey, suggesting that the disease is expanding and becoming more serious, i.e., in the provinces of Isabela, Eastern Samar, Leyte, Misamis Oriental, and Cotabato. As a trend, the CWBD incidence in the Philippines mostly occurred during March and April, with an average percentage of 5.6% of surveyed sites during March 2022 (Figure 3). In contrast, the disease incidence was lower from August and September to December, suggesting that symptom expression occurred more in the dry season than in the rainy season.

### 3.2. Phytoplasma Detection Using Nested PCR

Nested PCR targeting the 16S rRNA gene was performed on field-collected samples of symptomatic cassava from Isabela and Bukidnon provinces (Table 1). Positive reactions were observed in 6 out of 39 leaf samples from CWBD-symptomatic cassava, having an overall detection rate of 15.4% (Table 1). Target bands of approximately 1400 bp were detected in positive samples. Positive controls displayed clear single bands but not negative controls (Figure 4A).

In order to confirm the identity of the amplicons generated by nested PCR, *Sca*I restriction enzyme digestion and Sanger sequencing were used. For the 19 single bands generated by nested PCR, only 6 were confirmed as phytoplasma, producing two distinct bands of about 760 bp and 630 bp (Table 1, Figure 4B). Three representative samples are shown in Figure 4B (Lanes 1–3). Sequence analysis revealed that the *Sca*I-digested fragments were confirmed to be phytoplasma. Phytoplasma detected from cassava is closely related to *Ca*. P. luffae, belonging to group 16SrVIII-A, confirms the results of a recent study by Dolores et al. (2023) [25].

Despite the samples tested being in the advanced disease stage and showing distinct CWBD symptoms, the majority of them tested negative for phytoplasma, having a (false) negative rate of 80–89.5%. Most of the CWBD-symptomatic cassava leaf samples exhibited negative (below detection limit) and/or non-specific amplifications, accounting for over half or nearly all samples (Figure 4A; Supplementary Information). The method was also found to be highly prone to generating multiple bands and non-specific reactions, amplifying other bacteria other than phytoplasma. Two single-band amplicons not digested by *Sca*I were identified as various uncultured bacteria, primarily *Bacillus* sp. by sequencing (Table 1 and Appendix A, Figure 4B). Phylogenetic analysis for sequences of phytoplasma and other bacteria obtained in this study are shown in Appendix A.

### 3.3. Metagenomic Sequencing Analysis of Symptomatic Cassava Samples

To analyze phytoplasma presence in CWBD-affected cassava plants, the NGS approach was taken due to the technical limitations of the PCR detection method for phytoplasma. Details of the samples used are outlined in Appendix A. The enriched microbial DNA extracted from cassava plants from Bukidnon (A-1 to A-4) or Isabela (B-1 and B-2) were analyzed using 16S amplicon sequencing, which revealed bacterial community compositions in the cassava leaves showing typical CWBD symptoms (Figure 5). Employing the QIIME2 pipeline, each sequence read was assigned to 443 ASV identification numbers. The majority of reads were identified as chloroplast and mitochondrial sequences from host plants (Appendix A). Bacterial reads constituted 0.2% to 6.1%, indicating low bacterial colonization in cassava leaves (Figure 5). Phytoplasma reads were not present in 5 symptomatic cassava samples, while 1 sample (A-4) yielded phytoplasma reads that accounted for 9.2% and 0.6% of bacterial and total reads, respectively (Figure 5). Among cassava samples, other predominant genera were commonly included: i.e., *Pantoea* (3–50%), *Pseudomonas* (3–20%), *Methylobacterium*-*Methlorubrum* (3–34%), *Curtobacterium* (4–16%), *Sphingomonas* (2–11%), *Bacillus* (2–4%), and various uncultured bacteria (less than 33%) (Figure 5 and Appendix A). Additionally, genera such as *Exiguobacterium*, *Paenibacillus*, and *Staphylococcus* under the class Firmicutes were identified in low abundance (less than 6% of total bacterial reads) in cassava samples. Interestingly, *Ca*. Tremblaya, an obligate mealybug endosymbiont, was detected in cassava with CWBD (1.3–3.9% and 0.1% of bacterial and total reads, respectively). Most reads from all samples were assigned to genera under the classes *Bacilli*, *Alphaproteobacteria*, *Gammaproteobacteria*, *Actinobacteria*, and *Bacteroidia*, albeit with varying proportions among cassava samples (Figure 5 and Appendix A).

In order to survey the composition of other microbes in symptomatic cassava, non-targeted shotgun metagenomic sequencing was performed. Sequencing of microbial-enriched pooled symptomatic cassava DNA samples generated 100–121 million reads. Host filtering resulted in 22 million and 14 million unmapped reads for cassava (Appendix A), with overall alignment rates of 18.57%, and 7.38% for cassava samples CV-A and CV-B, respectively.

Taxonomic classification using Kaiju identified various taxa among genomic reads. Bacteria constituted 85.53–85.71%, followed by fungi (2.89–5.61%), and viral reads ranged from 0.28–0.32% of the total reads classified (Appendix A). *Streptomyces* was the most abundant bacterial genus (9.27–10.44% of bacterial reads), followed by, *Frankia*, *Pseudomonas*, *Vibrio*, *Xanthomonas*, *Sphingomonas*, *Microbacterium*, *Methylobacterium* and *Chryseobacterium.* Unclassified bacteria reads constituted almost a third of the bacterial reads (27.20–31.32%) while other bacterial genera comprised 38.14–42.97% of the bacterial reads. Reads identified as *Candidatus* Phytoplasma were also present in both samples but in extremely low abundance (0.001–0.003%), consistent with the 16S amplicon sequencing results (Appendix A).

In contrast to very low phytoplasma abundance, a significant number of reads from symptomatic cassava samples were assigned to the fungal genus, *Ceratobasidium* (39.72–52.63% and 1.14–2.94% of fungal and total reads, respectively) (Appendix A). Other fungal reads were classified into genera such as *Rhizoctonia*, *Golovinomyces*, *Fusarium*, *Valsa*, *Plectosphaerella*, *Cercospora*, *Rhizopus*, *Boeremia*, and *Stagonosporopsis*.

Viral reads from CWBD-symptomatic cassava samples primarily consisted of unclassified Caudovirecetes (77.92–81.52% of viral reads), with other classified viral families including *Geminiviridae*, *Mitoviridae*, *Picobirnaviridae*, *Caulimoviridae*, and *Casjensviridae*. Cassava samples contained reads classified under *Caulimoviridae* (unclassified Badnavirus).

*De novo* assembly using MEGAHIT generated 309,964 and 199,721 contigs from CV-A and CV-B samples, respectively (Appendix A). However, constructing phytoplasma genomes proved challenging due to low coverage and short-read sequencing limitations. Instead, barcode regions such as 28S rRNA and ITS of the most abundant fungus were analyzed from samples CV-A and CV-B (see below).

### 3.4. Microbial Composition in Symptomatic and Asymptomatic Cassava as Determined by RNA-Seq

Comparative analyses on microbial and viral compositions using field-collected CWBD-symptomatic and asymptomatic cassava plants were performed by RNA sequencing to obtain a more comprehensive analysis of the cassava leaf metatranscriptome. HISAT2 host read filtering generated 1.2 million to 12.3 million unmapped reads, with alignment rates to cassava genomes ranging from 82.13% to 95.32% of total read counts (Appendix A).

Taxonomic classification using Kaiju revealed that 99.6% to 99.9% of unmapped reads were classified into various taxa (Table 2). Fungi were relatively abundant across samples, constituting 10.5–85.3% and 4.6–11.7% in symptomatic and asymptomatic samples, respectively, followed by bacteria (8.9–60.5% and 10.6–73.5% in symptomatic and asymptomatic samples, respectively), along with other eukaryotic microbes.

In 3-month-old symptomatic cassava (I-1, I-2, and I-3), fungal reads constituted most of the classified reads, comprising 81.5% to 85.3% of total reads. Conversely, bacterial reads were predominant in leaves of 1-month-old CWBD symptomatic cassava from Isabela (I-B) that propagated in a screen house, and in field-collected asymptomatic 3-month-old cassava samples (H-1, H-2, and H-3), amounting to 60.5% and 57.9–73.5% of classified reads.

Alpha diversity was analyzed using the Shannon and Evenness indexes. The use of these indexes highlighted a significant difference in fungal diversity between symptomatic and asymptomatic samples (Figure 6). Symptomatic plants exhibited significantly lower fungal diversity compared to asymptomatic counterparts (*p* = 0.0002), with a mean Shannon index value of 3.2 for CWBD-symptomatic samples and 5.1 for asymptomatic samples. On the other hand, no significant differences were observed in bacterial (*p* = 0.624) and viral (*p* = 0.096) diversities between the sample types.

Among the classified fungal reads, the most relatively abundant and significant genus in symptomatic cassava was *Ceratobasidium*, comprising 26.1% to 59.5% of total fungal reads (2.8–41.2% of total reads) with a *p*-value of 0.018, consistent with shotgun metagenomic sequencing results (Appendix A), despite using different plant materials from Bukidnon and Isabela. Another significantly abundant fungal genus in symptomatic cassava was *Rhizoctonia* (5.3–12.6%) with a *p*-value of 0.018. The genus *Grifola* was abundant (29.0–31.2%) but only present in 3-month field-collected cassava, I-1, I-2, and I-3 and the difference was not statistically significant (*p*-value = 0.117). In contrast, CWBD-asymptomatic cassava contains a negligible percentage of the *Ceratobasidium*-classified reads (less than 0.5% and 0.0% of fungal and total reads, respectively) but have significantly abundant reads classified as genus *Valsa* (*p*-value = 0.018). Other fungal genera constitute 6.2–35.4% of symptomatic cassava reads, while 66.5–71.9% of asymptomatic cassava reads of fungi. Unclassified fungal reads make up 5.5–8.7% and 11.7–17.3% of total fungal reads from symptomatic and asymptomatic samples, respectively.

In terms of virus classification, most of the viral reads from symptomatic cassava were assigned to the family *Mitoviridae* (27.4–81.9% of viral reads; *p*-value = 0.018) (Figure 7). *Mitoviridae* comprises positive-strand RNA viruses, primarily infecting fungal hosts. Since viral reads were extremely low relative to total reads (0.2–1.7%), they are expected to originate from fungi infecting symptomatic cassava tissues. In asymptomatic plants, the significantly abundant viral reads were classified as *Caulimoviridae*, a family of dsDNA viruses known to infect plants (less than 0.3–14.8% of viral reads) followed by *Partitiviridae* (2.8–17.2% of viral reads; *p*-value = 0.018), a family of dsRNA viruses that infect mostly plants, fungi, and protozoa. Most of the other reads were classified as bacteriophages or viruses found in environmental samples (14.2–27.3% of viral reads). These combined data suggested that RNA viruses are unlikely to be causal agents of CWBD.

A similar trend to the 16S amplicon and shotgun metagenomic sequencing was observed in bacterial reads classification of microbial RNA-Seq reads. Among bacterial reads detected in cassava samples, the most abundant were from the genus *Pseudomonas* (Appendix A). Other abundant bacterial genera included *Pantoea*, *Sphingomonas*, and *Xanthomonas*. However, no phytoplasma reads were identified in either symptomatic or asymptomatic cassava samples (Appendix A).

### 3.5. Ceratobasidium sp. Detection by PCR and Sequence Analysis

Among the microbial taxa detected in symptomatic cassava using shotgun sequencing and microbial RNA-Seq, the most significantly abundant in CWBD-symptomatic plants were the fungal genus *Ceratobasidium*, the viral family *Mitoviridae*, and the bacterial genus *Pseudomonas*. Among these taxa, *Ceratobasidium* was previously reported as a plant pathogen that can cause similar symptoms in other crops, such as dieback, and vascular necrosis [64]. In checking the potential presence of this fungus with CWBD, amplification of the fungal barcode region ITS using ITS5 and ITS4 primers generated single bands with a length of around 700 bp in symptomatic plants. Sequencing of these bands and BLAST search found them to be highly similar to *Ceratobasidium* sp. in symptomatic plants. In asymptomatic plants, ITS primers did not produce bands or amplify the plant hosts ITS sequence. Similar results were obtained when amplifying the 28S rRNA gene of the fungus. On the other hand, no amplification was observed in asymptomatic samples for both target regions (Figure 8A,B). The generated phylogenetic tree showed that the sequences formed a separate clade from other *Ceratobasidium* and *Rhizoctonia* isolates and are highly similar to *Ceratobasidium theobromae* Sulawesi and South Sulawesi isolates and 22VDACS-RT12 isolate (accession numbers HQ424241, HQ424242, KU255724, HQ424246, and OQ361384) (Figure 9).

Detection of both phytoplasma and *Ceratobasidium* sp. in different tissues (i.e., leaves stems, and roots) of CWBD-symptomatic cassava yielded contrasting results. All of the symptomatic samples tested were found to be negative for the nested PCR targeting phytoplasma 16S rRNA gene for both 7-month and 9-month crop stages (Table 3). Most of the samples generated non-specific bands, but almost half of the samples produced single bands of the expected target size, which were found to not be from phytoplasma, based on *Sca*I digestion. On the other hand, *Ceratobasidium* sp. was detected in most of the symptomatic tissues using the specific primer designed to amplify the 28S rRNA gene. The expected band size of 748 bp was observed in all symptomatic plants and randomly selected amplicons were confirmed to be *Ceratobasidium* by sequencing. Positive detection of *Ceratobasidium* sp. can be observed in both pre-harvesting and harvesting stages and all tissue types such as leaves, stems, and roots (Table 3). Leaf tissues had the most positive results (96%) followed by stem tissues (88%).

Detection of both phytoplasma and *Ceratobasidium* sp. using the same sample was observed from the stem of two CWBD-symptomatic 9-month-old cassava (Table 3 and Appendix A). Using the specific primer for *Ceratobasidium*, 6 symptomatic leaves from the field-collected cassava positive with phytoplasma also tested positive for *Ceratobasidium* (Table 1, Supplementary Information).

Sequencing of randomly selected amplicons all produced sequences with 99.86% to 100% pairwise identity with *C. theobromae* South Sulawesi 6 and 22VDACS-RT12 isolates from Indonesia and USA, respectively (Accession No. KU319577 and OQ361384). Non-specific bands were not observed using *Ceratobasidium*-specific primer (Supplementary Information).

On the other hand, all CWBD-asymptomatic leaf samples were negative for both phytoplasma and *Ceratobasidium* sp. Similar results such as non-specific bands were observed from nested PCR targeting phytoplasma 16S rDNA. Some single bands were generated, however, *Sca*I restriction enzyme digestion and Sanger sequencing revealed that the amplicons are not from phytoplasma but from other bacteria (Table 3 and Appendix A).

## 4. Discussion

### 4.1. Current Status of CWBD in the Philippines and Southeast Asia

In this study, we investigated the status of the CWBD in the Philippines. Field observations and data obtained from the national authorities showed that CWBD is widespread in major cassava-growing areas in the Philippines and continues to cause severe crop damage that reduces yield and hampers the industry’s expansion (Figure 1, Figure 2 and Figure 3). CWBD still exists in the field despite the efforts of the national authorities and research institutions to manage CWBD since its reported occurrence in 2012 [14]. The government has implemented strategies, such as border control, information dissemination, and the application of streptomycin treatment targeting phytoplasma [74,75,76,77]. Aside from these measures, research has been conducted, particularly on the detection of a potential causal pathogen, surveillance and monitoring systems, the development of microbial inoculants as an alternative treatment for CWBD (biological control), and breeding of CWBD-resistant varieties [15,25,78,79,80,81,82].

Other countries in Southeast Asia, particularly Vietnam, Cambodia, and Laos, are also experiencing problems with CWBD [32,34]. CWBD symptoms reported in these countries are similar to those observed in the Philippines: stunting, witches’ broom, leaf chlorosis, dieback, and vascular necrosis (Figure 1). Advanced CWBD symptoms such as vascular streaking and leaf senescence, noticeable in the Philippines and Mainland Southeast Asia, were not apparent in previous accounts of the disease (Figure 1B,C). The unique vascular streak and dieback symptoms of CWBD in the Philippines are more similar to the typical symptoms of vascular streak dieback disease (VSD) found in cacao, which is caused by *Ceratobasidium theobromae* infection. Reassessing CWBD-infested plants in South America with *Ceratobasidium* sp. and phytoplasma detection is of great interest.

The peak of disease incidence was observed during the dry season of the Philippines, specifically in March and April (Figure 1). Interestingly, the peak of disease incidence during the dry season resembles the seasonality of VSD incidence. In cacao, VSD symptoms are more severe during drier periods. This was hypothesized to be caused by the impaired water transport in the xylem as a result of *C. theobromae* colonization. Before symptom expression in the dry season, very wet conditions during the rainy season favor sporulation and transmission of *C. theobromae.* Thus, incidence and severity are highly dependent on the rainfall patterns [83,84,85]. The similar trend in the infection dynamics of VSD and CWBD in the Philippines, including effects of the environment on pathogen dispersal, and the role of potential vectors if any, should be further investigated.

### 4.2. Low Detection Rate of Phytoplasma in Symptomatic Cassava by Nested PCR

In this study, we thoroughly investigated the potential presence of phytoplasma with CWBD. Nested PCR targeting partial sequences of genes encoding 16S rRNA, was found to be not reliable in detecting phytoplasma from CWBD-symptomatic cassava due to a high number of (false) negative results and non-specific amplification. Only 6 out of 39 cassava leaf samples were positive, as verified by *Sca*I digestion and sequence analysis of the PCR fragments (Table 1 and Appendix A). The result is consistent with the recent report of non-detection of phytoplasma in CWBD-infected samples in mainland Southeast Asia [32,34]. However, it contradicts a previous study conducted to identify a phytoplasma strain present in CWBD in the Philippines using a 16S rRNA gene that showed the detection of *Ca*. P. luffae (16SrVIII-A group) from symptomatic cassava leaves in the Philippines in 2017–2019 [25].

The conflicting findings can be attributed to the previous assumption that CWBD is caused by phytoplasma. Using only nested PCR designed to target phytoplasma, the previous understanding of the CWBD’s causal agent or associated pathogen is biased. There is no study yet on the etiology of CWBD by phytoplasma, nor any reports of extensive association studies with CWBD and phytoplasma. In some cases, phytoplasma detected in cassava may be a case of natural infection, as in the case of *Ca*. P. pruni in asymptomatic cassava in Japan [55]. Non-specific amplification and non-detection of phytoplasma by nested PCR may be attributed to its absence in most CWBD-infected samples in the Philippines (this study), Vietnam, Lao, and Cambodia [34]. This result confirms the previous findings that the existing PCR methods to detect phytoplasma are sensitive but prone to non-specific amplification [86]. However, the possible involvement of phytoplasma in CWBD was not fully excluded due to the unknown sensitivity of nested PCR, i.e., phytoplasma in non-PCR-positive samples does not indicate an absence of phytoplasma but rather possible presence below the detection limit. Thus, the verification of the diagnosis method by considering not only false positives but also false negatives may be important.

In addition, a recent study reported the existence of chimeric sequences in published 16S rDNA phytoplasma sequences, leading to incorrect assignment of strains [87]. Thus, the universal PCR detection method based on 16S rDNA must be used cautiously, and this is highly recommended to confirm by multiple molecular markers, other methods such as genome sequencing to classify phytoplasma, and use software to check for chimeric sequences [88,89]. To address these limitations, some studies have designed more specific primers to amplify 16S rDNA and sequenced phytoplasma genomes [90,91,92]. However, as is the case with CWBD, potential challenges may arise due to multiple phytoplasma species detected with a single disease, and extremely low accumulation of phytoplasma resulted in poor genome assembly.

These findings indicate that it is difficult to associate phytoplasma with CWBD, and focusing the detection solely on phytoplasma and assuming it is the causal agent without enough etiological evidence has skewed our understanding of the disease. Still, it is uncertain whether various phytoplasma detected in CWBD symptomatic plants are one of the potential causal agents of CWBD or even associated with the disease due to the absence of studies confirming the phytoplasma causality.

### 4.3. Microbial Composition in Symptomatic and Asymptomatic Cassava as Revealed by Next-Generation Sequencing

Next-generation sequencing techniques such as amplicon sequencing further confirmed the few positive detections with the absence or low frequency of phytoplasma reads from symptomatic cassava (Figure 4 and Appendix A). This finding is consistent with the low frequency of phytoplasma detected in highly symptomatic plants which was also observed in the lavender decline caused by *Ca.* P. solani [93]. In cassava, the most abundant bacterial genera in symptomatic cassava are *Pantoea* and *Methylobacterium*, but these are reported endosymbionts in cassava leaves and are not known to cause witches’ broom symptoms [94].

In contrast, non-targeted shotgun sequencing and microbial RNA-Seq revealed a relatively high abundance and unique presence of a fungus in the genus *Ceratobasidium* in CWBD-symptomatic but not in asymptomatic cassava (Figure 7 and Appendix A). *Rhizoctonia* sp. was another abundant fungus and known plant pathogen, but it was reported to cause cassava leaf blight disease and not witches’ broom disease [95]. Note that *Ceratobasidium* is a binuclear *Rhizoctonia* group genus, but the *Rhizoctonia* species discussed here refer to uninuclear and multinuclear *Rhizoctonia* groups [66]. A non-pathogenic fungus belonging to the genus *Grifola* was also abundant only in field-collected cassava plants, indicating a possible opportunistic infection of the endophyte presented in field conditions. Among the abundant fungal entities, *Ceratobasidium* members are more likely to be associated with CWBD due to their consistent abundance in multiple CWBD-symptomatic samples from Isabela and Bukidnon provinces, Philippines. Although some members of *Ceratobasidium* are known mycorrhizal symbionts, some are plant pathogenic and reported to cause similar symptoms observed in CWBD such as vascular necrosis [66,96,97,98].

Bacterial reads dominate the classified total read counts generated from shotgun metagenomics and metatranscriptomic sequencing (Table 2 and Appendix A). A higher percentage of bacterial reads can be observed in asymptomatic samples. The result may be explained by the higher diversity of bacteria in healthy plants' fungal reads, where a single genus (*Ceratobasidium)* dominates the proportion of read counts in symptomatic cassava. The rates of bacterial reads were significantly decreased in symptomatic plants (3-month-old) and were relatively affected in younger plants (1-month-old) because of massive colonization as indicated by a lower Shannon index (Figure 6).

For viral sequences, most of the reads from cassava belonged to the family *Mitoviridae* and unclassified bacteriophages from environmental samples such as uncultured cyanophages (Figure 7). Viral contigs are mostly unclassified Caudoviricetes, a class of viruses comprised of tailed bacteriophages that are abundant in nature [99]. Most of the classified viruses under the family *Mitoviridae* are mycoviruses [100,101]. There are no reports of these viral detections in cassava, but they were previously reported from the fungus *Ceratobasidium* and *Rhizoctonia* [102,103]. The abundance of the fungal genus *Ceratobasidium* and the virus family *Mitoviridae* follows a similar trend, indicating a possible association between the fungi and mycovirus. (Figure 7; compare those ratios in symptomatic and asymptomatic plants). However, the presence of mycoviruses in *Ceratobasidium* and *Rhizoctonia* and their role in infection must be validated by isolating the fungi, virus detection, and their characterization.

The result of alpha diversity analysis showed that only fungal communities have significant differences in diversity compared to bacterial and viral communities. The result is indicative of the lower diversity in symptomatic samples, maybe due to the dominance and colonization of *Ceratobasidium* in the diseased leaf tissues. In other studies, a lower Shannon index was observed in infected samples [104]. In addition, this confirms the result in Southeast Asia that CWBD-infected cassava has lower alpha diversity than healthy [34].

In some cases, other fungal and viral pathogens have also been reported to cause witches’ broom symptoms in other crops. The basidiomycete *Moniliophthora perniciosa* is known to induce witches’ broom in cacao [105,106]. In other studies, NGS analysis confirmed the association of a fungal pathogen, *Cophinforma mamani* in witches broom disease-affected *Nerium indicum* [107]. *Longan witches’ broom-associated virus* (LWBD), an ssRNA virus belonging to the family *Potyviridae,* is associated with witches’ broom disease in longan (*Dimocarpus longan*) [40, 41]. However, the results of NGSs conducted in this study possibly exclude viruses, phytoplasma, and bacteria in the symptom expression, but highlighted the predominant infection of *Ceratobasidium* sp.

### 4.4. High Detection Rate of Ceratobasidium and Rare Co-Infection Event with Phytoplasma

PCR amplification and sequence analysis of fungal ITS and 28S rDNA barcode regions indicated a very high similarity of CWBD-associated fungus with *Ceratobasidium* sp. (Figure 8 and Figure 9). Positive detection of *Ceratobasidium* sp. on infected plants further confirms the NGS result and highlights the possible connection with CWBD. *Ceratobasidium* sp. is a fastidious and obligate Basidiomycete fungus that resides on the xylem of tissues and typically infects cacao causing VSD [66]. The sequences found in cassava were identical to the *C. theobromae* isolate South Sulawesi 6, known to cause VSD disease of cacao in Indonesia and *C. theobromae* isolate 22VDACS-RT12 from VSD-infected redbud (*Cercis canadensis)* in USA. It is believed that *C. theobromae* is an endemic pathogen in South and Southeast Asia and transferred to cacao from an unknown original host [83]. Most importantly, ITS sequences obtained from CWBD-associated *Ceratobasidium* sp. in Vietnam, Cambodia, and Lao [34] were 98.5–98.8% similar to those obtained in this study and to the *C. theobromae* from VSD-infected cacao in Indonesia, suggesting that the same fungi can also be detected in CWBD in the Philippines.

Detailed observation of CWBD symptoms, particularly the prominent dieback during advanced stages of infection (Figure 1B) and the presence of brown streaking on vascular tissues (Figure 1D, E) indicate striking similarities with VSD symptoms associated with *C. theobromae* reported in other hosts [66,108,109]. Such vascular discoloration was also observed in the CWBD-affected cassava associated with *Ceratobasidium* sp. [34]. Browning of the roots is noticeably located in the xylem vessels and bundles (Figure 1E), adding to the possibility that the pathogen causing the vascular necrotic symptoms in CWBD is not phloem limited.

Simultaneous PCR detection of *Ceratobasidium* sp. and phytoplasma using the same samples from different cassava plant parts also indicated the presence of *Ceratobasidium* sp. with CWBD (Table 3). It was observed that leaves, roots, and stems from CWBD-symptomatic cassava harbor *Ceratobasidium* sp., have a high percentage of positive detection (69% to 96%). In contrast, phytoplasma is almost undetectable (0.8%) in all samples despite showing severe CWBD symptoms, similar to the non-detection of phytoplasma by a recent study [32]. It is noteworthy that the phytoplasma-positive cassava found in Japan did not show the CWBD symptom [55], and that it was free from a *Ceratobasidium* sp. infection (Figure 8B). In CWBD-asymptomatic samples, phytoplasma and *Ceratobasidium* sp. were not detected in all of the samples tested.

## 5. Conclusions

This study provides a comprehensive overview of the current status of cassava witches’ broom disease (CWBD) in the Philippines, utilizing data from secondary surveillance and primary field observations spanning 2020 to 2022. It underscores the persistent prevalence of CWBD across major cassava-producing regions nationwide. Despite exhaustive efforts to detect phytoplasma, molecular analyses revealed a low phytoplasma detection rate, instead uncovering the predominant infection of the fungus *Ceratobasidium* sp. in CWBD-affected cassava tissues, confirming the recent findings in mainland Southeast Asia. Although *Ceratobasidium* sp. showed significant abundance and consistent detection in CWBD-symptomatic plants, further studies are warranted to establish its causality through Koch’s postulate and to gain insights into disease development. Ongoing efforts include surveying additional cassava areas for *Ceratobasidium* sp., investigating its mode of spread, association studies, genetic diversity, in-planta localization, and enhancing existing diagnostic tools. These efforts aim to improve the understanding of CWBD and enhance the effectiveness of disease management strategies.

## Figures and Tables

**Figure 1 biology-13-00522-f001:**
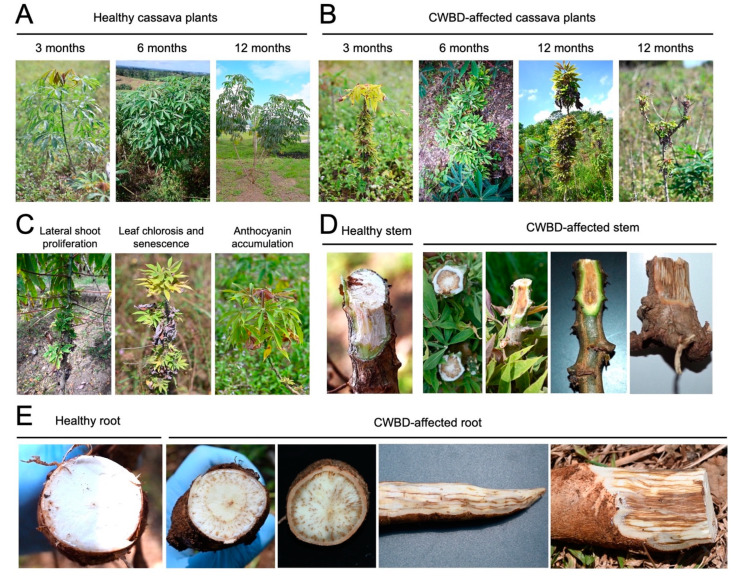
Symptoms observed in the CWBD-affected cassava fields in the Philippines. (**A**,**B**) Healthy or asymptomatic (**A**) and symptomatic (**B**) cassava plants at growing stages 3, 6, and 12 months. Typical witches’ broom symptoms were observed at each stage when a heavy incidence occurred. (**C**) Other representative symptoms of CWBD. (**D**,**E**) Browning of the stem (**D**) and root (**E**) tissues of CWBD-affected cassava. Highly symptomatic plants exhibited striated hyperpigmentation.

**Figure 2 biology-13-00522-f002:**
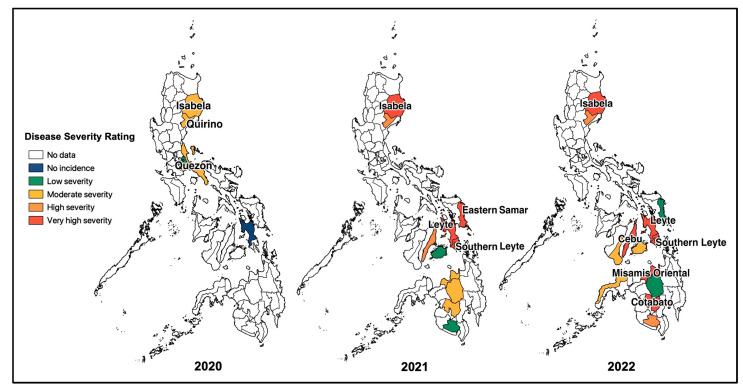
Map of the CWBD incidence and the highest severity in the surveyed major cassava-producing regions in the Philippines during 2020–2022. Blank areas are not targeted for the survey, while differently colored areas indicate the occurrence of CWBD with light to very severe symptom expressions (the most severe symptoms in the area were considered).

**Figure 3 biology-13-00522-f003:**
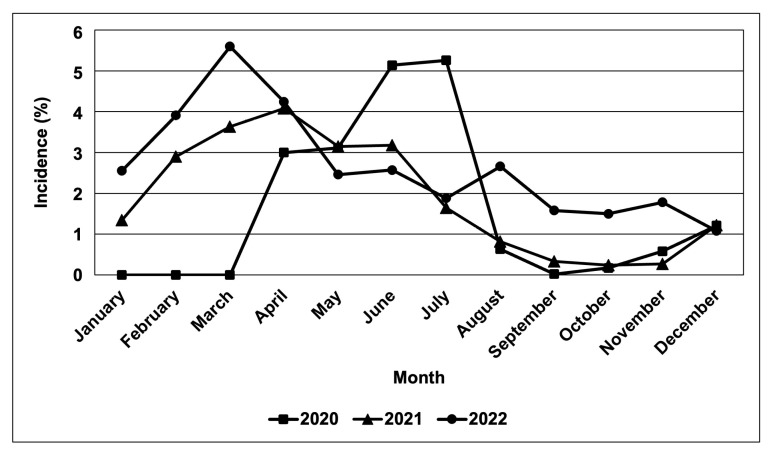
Rate of CWBD-occurrence in the surveyed sites. Monthly incidence was monitored in major cassava-producing regions in the Philippines during 2020–2022. The total averages of incidence rates are shown.

**Figure 4 biology-13-00522-f004:**
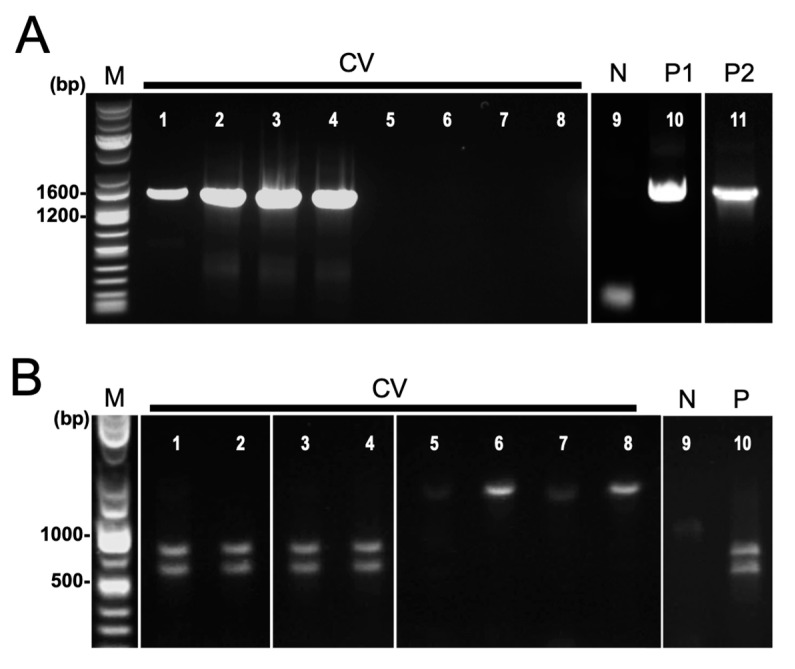
Detection of phytoplasma by nested PCR targeting 16S rRNA gene. (**A**) Agarose gel electrophoretic images of nested PCR amplicons using universal primer P1/P7 followed by R16mF2n/R1. Lanes: 1–8, field-collected symptomatic cassava leaves (CV); 9, negative control (N); 10, commercial positive control (P1, *Ca*. P. mali); 11, phytoplasma positive asymptomatic cassava (P2, *Ca*. P. pruni). (**B**) Restriction enzyme digestion of nested PCR products using *Sca*I. Lanes: 1–8, field-collected symptomatic cassava leaves (CV); 9, negative control (N); 10, commercial positive control (P, *Ca*. P. mali).

**Figure 5 biology-13-00522-f005:**
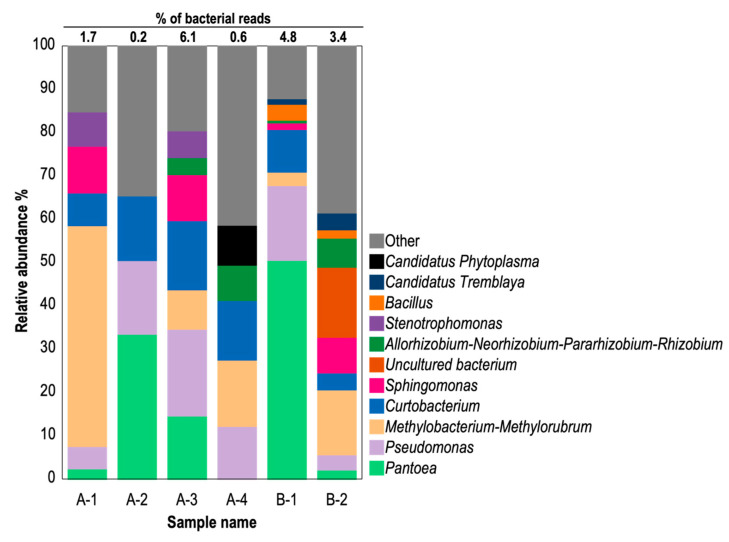
Relative abundance graph of 16S bacterial DNA amplicons in symptomatic cassava. Host-associated amplicon sequence variants (ASV) such as mitochondria and chloroplast were excluded in this analysis. Detected bacterial classifications (Genus-level) are color-differentiated. The percentage of bacterial reads against total reads, including those of the host, are indicated above the bar chart. A-1 to A-4, CWBD-symptomatic cassava from Bukidnon; B-1 and B-2, CWBD-symptomatic cassava from Isabela.

**Figure 6 biology-13-00522-f006:**
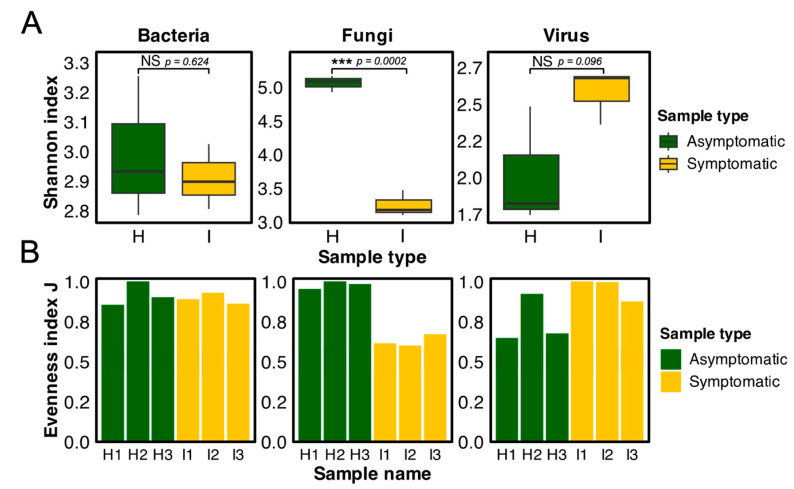
Alpha diversity analysis of bacterial, fungal, and viral communities in 3-month-old CWBD-symptomatic (yellow) and asymptomatic (green) plants. (**A**) Shannon index boxplots show significant differences in fungal diversity (*p* < 0.05, as denoted by ***) based on a t-test of independence while non-significant (NS) differences of bacterial and viral communities. (**B**) Evenness index bar plots showing normalized Shannon index of bacterial, fungal, and viral communities in symptomatic (I) versus asymptomatic (H) plants.

**Figure 7 biology-13-00522-f007:**
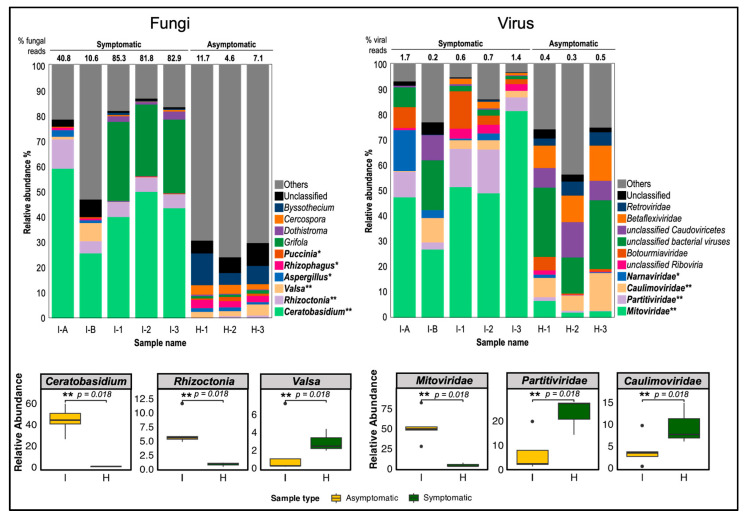
Relative abundance graph of fungal and viral host-filtered transcriptomic reads in symptomatic and asymptomatic cassava. Compositions (%) of fungal reads were classified by genus level while compositions (%) of viral reads were classified by family level. The most significantly different three taxa in fungi and viruses were statistically compared between diseased (I) and healthy (H) plant samples. Classified major taxa are color-differentiated and names are placed at the right of bar charts. Taxa with relative abundance significantly higher in symptomatic than asymptomatic cassava (*p* < 0.05) based on Mann Whitney U test are displayed in bold letter legend with asterisks (** *p*-value = 0.018; * *p*-value = 0.037). The percentage of fungal and viral reads against total reads is indicated above the bar chart. I-A and I-B, 1-month old CWBD symptomatic cassava; I-1, I-2, and I-3, 3-month-old CWBD-symptomatic cassava; H-1, H-2, and H-3, 3-month-old CWBD-asymptomatic cassava.

**Figure 8 biology-13-00522-f008:**
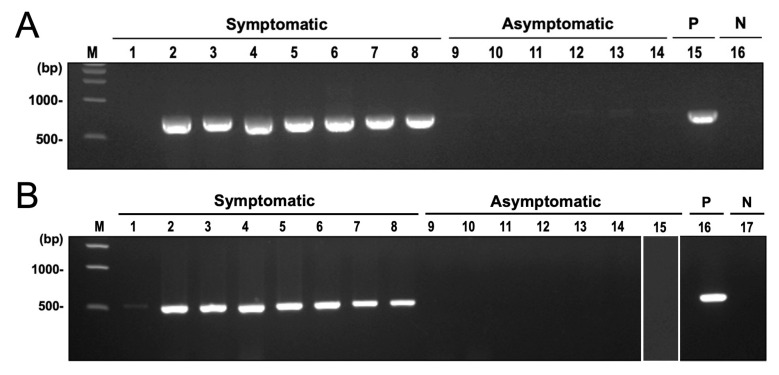
Fungal DNA amplification by end-point PCR in CWBD-symptomatic cassava leaves. (**A**) An electrophoretic migration of the PCR-amplified ITS fragments. Lanes: 1–8, field-collected symptomatic cassava samples; 9–14, field-collected asymptomatic cassava samples; 15, positive control (P); 16, negative control (N); M, 1 kb DNA ladder marker. (**B**) A *Ceratobasidium*-specific PCR-detection profile of 28S rDNA regions. Lanes: 1–8, field-collected symptomatic cassava samples; 9–14, field-collected asymptomatic cassava samples; 15, *Ca*. P. pruni-infected asymptomatic cassava sample from Japan; 16, positive control (P); 17, negative control (N); M, 1 kb DNA ladder marker.

**Figure 9 biology-13-00522-f009:**
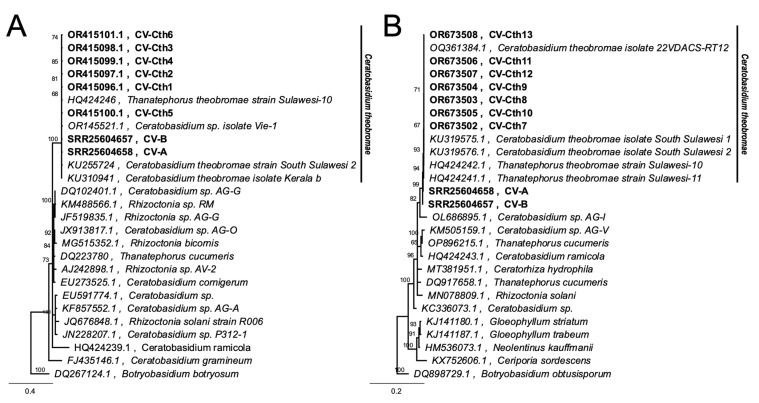
A Neighbor-Joining (NJ) phylogenetic tree of (**A**) ITS and (**B**) 28S barcode regions showing sequenced samples appear to be *Ceratobasidium* sp. CV-A, genomic contig from symptomatic cassava obtained in Bukidnon; CV-B, genomic contig from symptomatic cassava obtained in Isabela; CV-Cth1 to CV-Cth6, sequence from PCR amplification of ITS region of fungus from symptomatic cassava samples; CV-Cth7 to CV-Cth13, sequence from PCR amplification of 28S rDNA of *Ceratobasidium* sp. from symptomatic cassava samples. The tree was constructed using the Tamura-Nei genetic distance model with 1000 bootstraps and percentages at nodes. The scale bar indicates 0.2 and 0.06 substitution per site.

**Table 1 biology-13-00522-t001:** Detection of phytoplasma from symptomatic cassava leaves by nested PCR, and *Sca*I digestion of PCR amplicons.

Location	Stage/ Sample Type	Nested PCR ^1^	*Sca*I Digestion ^2^	Detection Rate	False Negative Rate	Classification ^3^	Clone Name	Accession Numbers
Isabela	Pre-harvesting (7 months) Leaves	8/19 (42%)	2/8 (25%)	2/19 (10.5%)	17/19 (89.5%)	*Ca*. P. luffae 16SrVIII-A	CV-Phy4 CV-Phy6	OR673513 OR673514
Bukidnon	Pre-harvesting (7 months) Leaves	11/20 (55%)	4/11 (36%)	4/20 (20%)	16/20 (80%)	*Ca*. P. luffae 16SrVIII-A	CV-Phy1 CV-Phy2 CV-Phy3 CV-Phy5	OQ797687.1 OQ797688.1 OR673512 OR673511
						Not phytoplasma (other bacteria)	CV-Ub10 CV-Ub11	OQ797685.1 OQ797686.1

^1^ No. of positive/no. of tested samples (percentage); ^2^
*Sca*I enzyme digestion performed on nested PCR-positive samples to verify phytoplasma sequences; ^3^ Species and 16Sr group/subgroup classification *i*PhyClassifier.

**Table 2 biology-13-00522-t002:** Classification of host-filtered transcriptomic reads using Kaiju.

	Symptomatic					Asymptomatic		
Stage	1-Month-Old ^1^		3-Month-Old ^2^					
Classification	I-A	I-B	I-1	I-2	I-3	H-1	H-2	H-3
Fungi	40.8%	10.6%	85.3%	81.8%	82.9%	11.7%	4.6%	7.0%
Bacteria	36.3%	60.5%	8.9%	11.4%	10.6%	58.1%	73.5%	57.9%
Viruses	1.7%	0.2%	0.6%	0.7%	1.4%	0.4%	0.3%	0.5%
Archaea	0.1%	0.1%	0.1%	0.1%	0.1%	0.3%	0.5%	0.3%
Others	20.9%	28.4%	5.0%	5.9%	4.9%	29.2%	20.8%	34%
Unclassified	0.4%	0.2%	0.1%	0.1%	0.1%	0.4%	0.3%	0.3%
Total classified	99.6%	99.8%	99.9%	99.9%	99.9%	99.6%	99.7%	99.7%

^1^ Leaves collected from screenhouse-propagated 1-month-old CWBD symptomatic cassava from Bukidnon (I-A) and Isabela (I-B); ^2^ Field-collected leaves from 3-month-old CWBD symptomatic cassava in Isabela.

**Table 3 biology-13-00522-t003:** Detection rate of phytoplasma and *Ceratobasidium* sp. in cassava.

	Phytoplasma 16S rDNA		*Ceratobasidium-*Specific 28S		Co-Infection
Sample Type	PCR/*Sca*I Digestion ^1^	Accession Numbers	PCR Result	Accession Numbers	PCR Result
Symptomatic Pre-harvesting (7 months)					
Leaves	0/105 (0%)	OQ797681.1 ^2^ OQ797682.1 ^2^ OQ797683.1 ^2^ OQ797684.1 ^2^	81/105 (77%)	OR673508.1 OR673503.1	0/105 (0%)
Harvesting (9 months)					
Leaves	0/23 (0%)	NT ^3^	23/23 (100%)	NT ^3^	0/23 (0%)
Roots	0/13 (0%)	NT ^3^	9/13 (69%)	OR673502.1	0/13 (0%)
Stem	2/97 (2.1%)	NT ^3^	91/97 (93.8%)	OR673504.1 OR673505.1 OR673506.1 OR673507.1	2/97 (2.1%)
Asymptomatic (9 months)					
Leaves	0/50 (0%)	NT ^3^	0/50 (0%)	NT ^3^	0/50 (0%)

^1^ *Sca*I enzyme digestion performed on some of the nested samples to verify phytoplasma sequence. ^2^ Sequenced amplicons of other bacteria (not phytoplasma). ^3^ Not tested.

## Data Availability

The accession numbers for sequence data obtained in this study are listed in Appendix A. Raw reads from 16S amplicon sequencing, shotgun metagenomics, and microbial RNA sequencing were deposited in the NCBI Sequence Read Archive BioProject numbers PRJNA956085, PRJNA1003250, PRJNA990661, and PRJNA1028178, respectively.

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
