# Peer review of "Status of Cassava Witches’ Broom Disease in the Philippines and Identification of Potential Pathogens by Metagenomic Analysis"

_biology, 2024, doi:10.3390/biology13070522_

Round 1

Reviewer 1 Report

Comments and Suggestions for Authors

The authors claimed an association of phytoplasms and Ceratobasidium theobromae (fungus) with cassava witches’ broom disease. Based on: a) PCR detection and sequence homology of phytoplasms, and b) higher relative abundance of sequences belonging to C. theobromae in microbiome of symtomatic cassava plants.

The plant-phytoplasm interaction is an intriguing and complex pathosystem, which raises many questions. Who is the causal agent? What is the virulence mechanism involved in plant symptomatology? And more… At the beginning, I was excited about the use of NGS to explore the microbiome/microbiota of symtomatic plants. However, I found several facts  that make me doubt about the main conclusion “association of phytoplasms and Ceratobasidium theobromae with cassava witches’ broom disease”, since:   1)  No detection of phytoplasma in symptomatic plants though PCR, would be due to low levels of DNA target. The PCR requires a minimal DNA amount to show amplification. In the case of manuscript, authors should discuss about how discard false negative, as a result of low levels of DNA target. 2)  The manuscript pointed out results regarding to BLAST [Table S4](amplicon size, Evalue, %cover, %identity) and phylogenetic reconstruction [line 465] (gene size, alignment algorithm, nucleotide substitution model, phylogenetic algorith, Bootstrapp), however related information is missing. For instance the phylogenetic tree based on cpn60 and secA partial gene sequences is missing. 3)  In other hand, 16S metabarcoding, could allow to explore the taxonomic diversity of bacterial microbiota of healthy and infected cassava plants, with ecological measures + statistical tests. 4)  Regarding to shoutgun metagenomics and RNA-seq, could allow to explore both taxonomic and functional diversity, in other words the microbiome of healthy and infected cassava plants, with ecological measures + statistical tests.   5)  However, instead manuscript focuses on relative abundance of particular fungal sequence without statistical test. Therefore, I perceive that hypothesis of manuscript is not supported, due to to lack of statistical tests. 6)  In general terms, the manuscript is complex to follow up. Too many figures and tables. My recommendation is to synthesize even more and limit the different sections of results. For instance “PCR detection section”, “16S metabarcoding section”; “shoutgun metagenomics section”, “RNA-seq section”. Every section should end with the main contribution and explanation to continue with the next approach, which would confirm/wide the results 7)  Final comment: the cassava-phytoplams interaction is interesting and intriguing also very complex and heterogeneous. The application of omics-approaches is understood, but it is desirable to make it in integrative manner, to exploit the taxonomic and functional diversity of plant microbiome under specific circunstances. I suggest the authors reconsider the approach to exploring the microbiome of healthy and diseased plants, either by 16S metabarcoding, shoutgun metagenomics or RNA-seq or in an integrative context. 8)  Below are some papers related to exploration of plant microbiome

https://doi.org/10.1186/s40168-021-01138-2

https://doi.org/10.1094/PBIOMES-11-22-0091-R

Below are point comments:

TITLE: “…with cassava witches’ broom disease in Philippines”.

ABSTRACT

Line 29: “… in Philippines in 2012, CWBD…” AND SO ON..

Line 31: “…was detected in CWBD-infected cassava plants in Philippines in 2017, and it was considered as the causal agent, despite a lack of evidence regarding etiology and transmisión…”

Line 33: Most recently, asociation of (FUNGAL FULL NAME)  with the CWBD has been reported in Vietnam.

INTRODUCTION

Line 73: “…phytoplasma, an obligated phytopathogen phloem-colonizing bacteria lacking a cell wall [12–14]. Phytoplasma is…”

Line 102: “…to perform culture-independent, high-throughput profiling…”

Lines 107-111: The paragraph is not comprehensible.

MATERIALS AND METHODS

Line 125: Photographs belonging to Fig. S1 are blur.

Line 132: “…extraction and propagation in the  greenhouse with permission from local authorities…” AND SO ON.

Fig. S2: Greenhouse-propagated 1 month-old cassava with witches’ broom symptom used for microbial RNA-Seq.

Line 308: Authors should indicate: a) alignment algorithm, b) why Tamura-Nei distance model was selected as nucleotide substitution model. b) bootstrapp permutations

Line 314: “…16S amplicon sequencing data…” Authors should indicate diversity parameters employed in this study (rarefaction curves, a-diversity, b-diversity …), to address the ecological side.

Line 387: BLAST results of sequences from leafhoppers are missing (size amplicon, Evalue, cover, identity, Accession, picture of espcimen).

Line 389: Fig 4ª revealed non-specific PCR amplification in weed samples. The RFLP did not clarify. Authors should clarify the sentence.

Line 396: BLAST results of sequences are missing (size amplicon, Evalue, cover, identity, Accession).

Table S3 did not correspond to sequence results of COI insect

Line 433: “…endpoint PCR amplification of partial sequence of cpn60 and secA genes were initially…” The respective amplicon size is missing for each gene sequence.

Line 440: Authors should indicate the process to design specific primers for cpn60 gene. Also, physicochemical caracteristics and sequence of designed primers. It is not clear the need for designing such primers. The respective amplicon size is missing for each gene sequence.

Line 465: Phylogenetic reconstruction of cpn60 and 16S rDNA partial sequences is missing.

Line 486: The authors did not clarify if the 16S rRNA amplicon sequencing was conducted with sample replicates at least n=3.  A description of Fig. S6 is missing.

Line 496: “Only one of six tested cassava samples (CV-A4) yielded phytoplasma reads that accounted for 9.2% and 0.1% of bacterial and total reads, respectively (Figures 7, 8, S3, and 497

S4)”. The figures did not correspond to sentence. The sample CV-A4 holded aprox 10% (relative abundance of Phytoplasms sequences.

è Regarding to 16S metabarcoding, I strongly recommend to authors to conduct a description of bacterial/fungal/viral microbiota based on alpha/beta diversity, clustering, correlation, enrichment analysis, etc. In order to reveal the taxonomic structure of respective microbiota from a ecological point of view, also to reveal biomarkers taxa with statistical significance. Which could drive to biological signficance.

è Regarding to NGS shotgun analysis, the term should be microbioma. The concept microbiome includes taxonomic structure and the repertoire of genes found. So, the analysis of microbiome of phytoplams-infected cassava plants should be included. Since the description of relative abundances is a poor description of a complex microbiome.

DISCUSSION

Line 683: the section contains information reviewed in Introduction section.

è Authors should discuss about the requirements for conventional PCR, like minimun DNA quantity required for PCR amplification. It is likely a very very low quantity of sequence target unable to perfom visible PCR amplification. It means low levels of phytoplasm infection.

Line 71): “…Limited presence of phytoplasma in CWBD-affected tissues using conventional approaches”. What means "limited presence"?A quantitative approach should be included to avoid subjetivity.

Line 721: “…Nested PCR targeting partial sequences of genes encoding 16S rRNA, cpn60, and secA…”

è Authors should discuss about the requirements for conventional PCR, like minimun DNA quantity required for PCR amplification. It is likely a very very low quantity of sequence target unable to perfom visible PCR amplification. It means low levels of phytoplasm infection.

Line 801: “…unclear correlation between the disease and phytoplasma or bacteria but a strong association with the fungus, Ceratobasidium sp….”

è The correlation analysis is missing.

Line 808: “…Positive detection of C. theobromae further confirms the NGS result and strengthens fungal association with CWBD”.

è NGS possess a higher sensivity than conventional PCR. It is obvious and notorious…

Comments on the Quality of English Language

 Minor editing of English language required

Reviewer 2 Report

Comments and Suggestions for Authors

The manuscript represents a very interesting work on association of diseases to possible different pathogens. Cassava witches’ broom disease (CWBD) is one of the most devastating diseases of cassava, and it threatens global production of the crop. While it is known worldwide that cassava witches’ broom disease is mainly associated to phytoplasma presence, the authors clearly showed that in plants with severe symptoms, particularly the prominent dieback during advanced stages of infection and the presence of brown streaking on vascular tissues, the phytoplasma titre was very low and only few samples resulted positive in PCR.

This manuscript highlights an interesting and complete study on the complexity of the disease and the authors try to deeply understand the possible causative agents in a very wide manner, not focusing only on what it was already published, but investigating other possible causes. Following NGS detection approach to verify the presence of bacteria, fungi and viruses in the analyzed samples the authors report a high abundance of fastidious and xylem-limited fungus, Ceratobasidium theobromae, that causes witches’ broom in another host and that can though associated to cassava symptoms. The experimental design and the methods applied are trustable and reproducible and the methidology is presented in details.

The results are organized in a comprehensive way and tables and figures, together with the supplementary materials, help readers to fully understand the section.

The conclusions are based on the results and the cited literature is appropriate and exhaustive.

Taking all in considerations I would recommend the publication of the manuscript in its present form.

Author Response

Thank you very much for analyzing our manuscript and understand what we focused on in this study. We will revise the manuscript based on other reviewers’ comments to further improve. 

Reviewer 3 Report

Comments and Suggestions for Authors

The main question addressed by this research is to identify a potential causal agent of cassava witches’ broom disease (CWBD) in the Philippines using PCR and NGS techniques. The authors presented results of the association of Ceratobasidium sp. with partial involvement of phytoplasma Ca. P. luffae and DNA/RNA viruses.

This topic is relevant in the field of plant pathology, taking into account the economic significance of cassava and occurrence of cassava witches’ broom disease in several countries that is threatening the production. The paper presents the results of the comprehensive study in order to reveal the cause of the disease.

The presented study is the first complete study of the disease through the analysis of the infected material for the presence of various plant pathogens (phytoplasmas, fungi, bacteria, viruses) as suspected causal agents of the disease. Previous reports were focused only on one pathogen, ignoring other causative agents or synergism between different organisms.

The methodology is presented in detail (somewhere in too many details), but it does not have to be a flaw in the manuscript. I do not have any complaints in this section.

Presented conclusions are consistent with results and given arguments answering the main asked question.

Listed references are appropriate and cited properly.

Figures and tables are illustrative and informative. I have the same opinion for supplementary files.

Author Response

Thank you very much for your careful analysis of our manuscript. We particularly appreciate your understanding of the values that we surveyed all potential pathogens (phytoplasma-bacteria, fungi, and viruses). We will revise the manuscript based on other reviewers’ comments to make it a much more comprehensive report.

Reviewer 4 Report

Comments and Suggestions for Authors

The work by Landicho et al. describes the association of cassava witches broom disease (CWBD) with Ceratobasidium theobromae and not with phytoplasma. The results are not novel but significant as a large group of previous works took for granted phytoplasma as the causal agent of CWBD. The results confirm a recent work showing that CWBD is rather associated with a fungus of the Ceratobasidium genus, including the analysis of symptomatic plants where phytoplasma was not detected. 

The authors have described a very interesting set of results from PCR tests in different tisuues for Phytoplasma and the fungus. I suggest they try to calculate the association between phytoplasma and Ceratobasidium, considering your results on PCR amplification (and confirmation by sequencing).

It is important to keep in mind that identifying the fungus to the species level based on ITS data is not conclusive. In the absence of additional genomic data, additional conserved markers such TEF1, RPB2, ATP6, are recommended to confirm the identity at species level. Until more data is available, it is recommended you modify your title and keep naming the fungus Ceratobasidium sp.

Additional suggested modifications to the text are listed below:

Line 34: Change “Vietnam” for “Lao PDR”.

Line 37: change “PCR-based detections of phytoplasma are…” for “PCR-based detection confirms that phytoplasma are…”

Line 76-77: It is already reported that the witches’ broom symptoms observed in Brazil are different from those known as CWBD in Southeast Asia. Please modify the text accordingly. Include also the report of cassava witches broom symptoms from Uganda (Arocha et al., 2009)

Line 77: Please add a primary reference indicating the presence of CWBD symptoms since the 1990s in Southeast Asia. Reference 11 only refers to works published in the 2000s.

Line 119: Please explain why you chose 50 plants per site.

Line 131: Please indicate if the leaves samples included the short petioles observed in plants with CWBD symptoms. Also include here information of how many samples per site were collected and later analyzed by PCR (out of the 50 observed per site).

Line 291: Did you normalize the data by sequencing depth before comparing the taxonomic composition of the samples? This step is important for comparative analysis and should be applied to all three strategies used: 16S, metagenomics and RNA-Seq.

Line 283: The platform is called Illumina® NovaSeq 6000, not Illumina® NovaSeq 60000. Please correct this.

Line 303: Maintain writing format: use BLASTn (as on line 294) or blastn throughout the text.

Line 315: Specify the version of QIIME2.

Line 315: Specify the quality control method used in the QIIME2 pipeline: DADA2 or Deblur.

Line 315: Specify the 16S database used for the QIIME2 classifier (Greengenes, SILVA, etc.).

Line 633: Please indicate if the expected 700 bp PCR band was obtained in ‘all’ symptomatic plants (and whether you confirmed all of them by sequencing). Table 4 should indicate if all the results described there are from symptomatic plants only.

Line 699-702: Move this information on distinct witches’ broom symptoms in the Americas versus Southeast Asia, to the introduction.

Line 728-729: Reference 27 did not evaluate the association between PCR and CWBD symptoms.

Line 806-844: Please note that in order to identify the fungus at the species level, you need to use additional markers. Conserved genes such as TEF1, RPB2, ATP6 are recommended in the absence of additional genome information. IN the absence of this evidence, it is recommended use Ceratobasidium sp. to refer to the fungus found associated to CWBD.

Comments on the Quality of English Language

The work is described clearly. Only minor corrections are needed.

Reviewer 5 Report

Comments and Suggestions for Authors

Using the next-generation sequencing (NGS) techniques, an association between the fungus C. theobromae and Cassava witches’ broom disease (CWBD), whih is one of the most devastating diseases was revealed by the authors, providing a potential cause of CWBD for further investigation. However, some issues need to be addressed properly.

(1)  Despite a possible association between C. theobromae and CWBD, additional validation is needed to establish its causality by Koch’s postulate, which is a necessary scientific step to make sure the right etiology and causal agent.

(2)  The abstract is too long, and it should be short and precise.

(3)  In line 283, “pair-end sequenced (2 x 150 bp)” and “pair-end sequenced (150bp×2) in line 287” should be unified as 150bp×2.

Comments on the Quality of English Language

Using the next-generation sequencing (NGS) techniques, an association between the fungus C. theobromae and Cassava witches’ broom disease (CWBD), whih is one of the most devastating diseases was revealed by the authors, providing a potential cause of CWBD for further investigation. However, some issues need to be addressed properly.

(1)  Despite a possible association between C. theobromae and CWBD, additional validation is needed to establish its causality by Koch’s postulate, which is a necessary scientific step to make sure the right etiology and causal agent.

(2)  The abstract is too long, and it should be short and precise.

(3)  In line 283, “pair-end sequenced (2 x 150 bp)” and “pair-end sequenced (150bp×2) in line 287” should be unified as 150bp×2.

Reviewer 6 Report

Comments and Suggestions for Authors

The manuscript addressed the Next-Generation Sequencing Reveals Limited Phytoplasma and Potential Role of Ceratobasidium theobromae in Cassava Witches' Broom Disease in the Philippines. The manuscript is generally good However, the following points need to be considered. 

The title should be changed to a more simple one such as 'Limited Phytoplasma Detected and Ceratobasidium theobromae Associated with Cassava Witches' Broom Disease in the Philippines Using Next-Generation Sequencing"

Abstract: Briefly discuss alternative explanations for low phytoplasma detection (e.g., uneven distribution within the plant, specific detection limitations).

Introduction: The section on phytoplasma detection (paragraphs 83-89) could be condensed slightly as some information overlaps with earlier parts (paragraphs 72-74). Focus on the limitations of phytoplasma detection in the context of CWBD.

The mention of Ceratobasidium sp. as a potential causal agent (paragraph 96) could be brought forward and elaborated on a bit more. This is a significant development and could be positioned as a potential shift in understanding CWBD etiology.

Materials and methods:

In terms of sample size, mentioning the number of cassava plants, insects, and bitter gourd/sponge gourd samples collected would be informative.

Briefly explaining why 1-month-old and 3-month-old cassava plants were chosen for RNA-Seq analysis would be helpful.

Please provide the rationale for de novo assembly in  "Genomic reads were de novo-assembled using MEGAHIT v1.2.9 to generate contigs'

Results:

Please make sure that all scientific names given in figures 6, 7 and 9 are italic.

Discussion: 

L209-217: While the xylem colonization explanation is compelling, briefly mentioning other factors potentially influencing seasonality (e.g., insect vector activity) could add nuance.

 Briefly acknowledge the rationale behind the previous study that detected phytoplasma (Ca. P. luffae) to add context.

While advocating for more specific primers and genome sequencing, please consider  the limitations and potential challenges associated with these approaches to provide a more balanced perspective.

l782-791:please specify the types of bacteriophages identified (e.g., plant- or insect-associated) ' unclassified Caudoviricetes" is informative, '

References:

Please make sure all scientific names are italic throughout the reference list.

Comments on the Quality of English Language

Minor editing of English language required

Round 2

Reviewer 4 Report

Comments and Suggestions for Authors

Thanks to the authors for addressing all the comments presented to them and adjust the text accordingly. The work contributes positively to understanding the disease known as cassava witches' broom disease.

Page 1, Line 3: In the title, please add "symptoms" after "witches' broom" 

Page 2, Line 63: Please refer here to the symptoms instead of the name of the disease. In Brazil these symptoms were described as Superbrotamento or Vassouramento, and comprised a group of symptoms, one of which were similar to those observed in CWB in Southeast Asia. You could indicate that: "Similar symptoms, described as Superbrotamento or Envassouramento, were observed in Brazil in the 1940s".   

The above comment is related to the statement in page 20, Line 761-762, where the authors cautiously indicate "it is believed that C. theobromae is an endemic pathogen in Southeast Asia". Please elaborate here alternative escenario in the case witches' broom symptoms were already observed in the Americas since the 1940s.

Page 2, Line 90: please change to: "...and the fungus Ceratobasidium sp. was discovered using a metagenomic approach and found associated with CWBD using a set of specific PCR primers [34]"   

Page 5, Line 28: Leiva et al., 2023. present a pair of PCR primers as a robust diagnostic tool for CWBD, based on association data). It would be interesting to show if this is still true for isolates of the fungus found in the Philippines. Please consider testing this with your samples.

Previous isolates of Ceratobasidium sp. from cassava showed ITS nucleotide identities higher than 98.3% to Ceratobasidium theobromae. In page 20, Line 764, the authors indicate that isolates from the Philippines showed identities higher than 98.5% with Ceratobasidium theobromae from cacao. However, they conclude the fungus from the Philippines has a different origin than those from mainland Southeast Asia. The relatedness of these isolates (cacao vs, cassava) using ITS sequences needs to be re-analyzed. The ITS tree presented in Figure 9 is also not conclusive. Does the 16S data support this statement? Otherwise, I would recommend withdrawing this statement from the Discusion.

Comments on the Quality of English Language

No specific comments.
